## PERSPECTIVE

# Sensor extended imaging workflow for creating fit for purpose models in basic and applied cell biology

Julia Schueler [1]✉, Heikki Sjöman [2] & Carlo Kriesi [2]

While various engineering disciplines spent years on developing methods and workflows to increase their R&D efficiency, the field of cell biology has seen limited evolution in the fundamental approaches to interact with living cells. Perturbations are mostly of chemical nature, and physiologically relevant contexts and stimuli are left with limited attention, resulting in a solution space constrained within the boundaries of presently manageable perturbations. To predict in the laboratory how a drug will work in a human patient, cell biology must have a closer look at life and strive to mimic the human being in all his complexity. By implementing an iterative process from perturbation to measurement and vice versa, the authors suggest using a sensor-extended imaging workflow to implement product development practices to cell biology, opening a physiologically relevant solution space for the development of truly translational and predictive fit for purpose in vitro cell models.

The recently passed FDA Modernization Act aims to integrate advanced in vitro models of different diseases into the drug development process. This legislation opens the door for alternative animal-free testing grounds not only in efficacy but also in toxicity studies[1]. The implementation of this legal framework mirrors the development in the field to mimic human disease not any longer in animals, mostly rodents, but rather develop in vitro systems aiming to reflect human disease more realistically using primary human tissue in sophisticated culture systems. With the first models successfully being commercialized, the question is *when*, not *if* these advanced disease models will broaden their application from basic research toward drug development. This development has several implications: (a) it is getting more and more cumbersome to make an informed decision on which model to use for what kind of application since, (b) an increasing number of read-outs and therefore data have to be aggregated, annotated, analyzed and set into the biological context, (c) the development of a bespoke assay needs to be a multidisciplinary approach as combined input from biology, data science, and engineering is critical for success.

This perspective is not trying to touch the technical aspects of integrated sensors for advanced cell models, such as Fuchs et al.[2] have already done, but it aims to outline how the adoption of workflows already successfully implemented in other scientific areas has the potential to unleash the full power of the innovative technologies currently evolving in the field of advanced 3D in vitro assays and beyond.

It is our ambition to kick off the discussion on the current workflow of product development in cell biology. At present, many research groups are stating that their work is translational and predictive, while the success rates of new drug programs remain below 20%[3]. Based on this, the authors argue that it is time for biology to expand the traditional approach where one observes the truth with an engineering approach that transforms the nature of the underlying experiments to interactive context-dependent design problems. Further borrowing from design science, when a task is too complex to plan everything in advance, we need to proceed iteratively toward the end goal. Based on the multidisciplinary background of the authors, this article summarizes how

[1] Charles River Germany GmbH, Am Flughafen 12-14, 79111 Freiburg, Germany. [2] Vitroscope AS, Leirfossvegen 5d, 7037 Trondheim, Norway.
✉email: julia.schueler@crl.com

other disciplines tackled similar crossroads in the development path and proposes to apply similar tactics for developing future advanced in vitro cell models.

**The promise and the crossroads of in vitro cell models.** Since the first experiments involving animal cells in isolation approximately 100 years ago, the medical need especially in virology and oncology has initiated the development of a plethora of different in vitro technologies that aim to mimic the human diseased or healthy phenotype[4]. The main applications as of today are disease modeling, precision medicine, and regenerative medicine. The advantage of in vitro models is the possibility to identify critical cellular and molecular contributors to the disease by permitting manipulation of each cell type in isolation[5]. There are several research areas to which cell culture systems are important contributors, such as (a) modeling specific diseases to expand our knowledge of the disease and ultimately develop a cure, (b) to understand the physiological requirements of bespoke cell types and along those lines be able to determine negative and positive effects of perturbations on those cells, (c) as basis of manufacturing systems for therapeutic proteins, cell, and gene therapies[6–9].

At each step of designing the actual assay, the scientist has multiple options for the workflow. Not only the cell source but as well the culture conditions, the nature of the perturbation, and the choice of read-out(s) influence the requirements. With the possibility to follow biological processes in a time-resolved manner the complexity and size of the data generated per experiment increased once more making a bioinformatic-supported analysis an inevitable necessity. This additional layer of complexity can be handled with the support of machine learning and artificial intelligence tools[10]. These enable the interpretation of the actual data but as well put the results into a larger context by integrating publicly available datasets (Fig. 1).

The cellular component of the assay is represented either by primary human cells or iPSC's in case the sourcing of the cells is extremely challenging like in the neuroscience space[11,12]. The source of the cells as well as accompanying metadata must be factored into the interpretation of the study results. The donor patient's history, such as response to pre-treatment, age, gender, ethnicity, and other phenotypical features must be brought into context with the assay read-outs. The architecture of the tissue is another crucial component to be mimicked in the in vitro assay. The invention of 3D models increased the translational value of in vitro cell models tremendously[13]. Nevertheless, our understanding of spatial dependencies is still fractured and can only be deconvoluted with the help of image analysis tools that enable the quantitative analysis of highly complex tissue aggregates[14]. Beyond the structural architecture of a tissue, physical and chemical stimuli do have an impact on the biological status of the cells. The modulation of these factors leads to different outcomes and therefore must be closely monitored. The effect of the pH for example can have a tremendous effect on the predictivity of the assay and can hamper a drug development pipeline substantially[15]. Other stimuli such as temperature and mechanical stress are currently mainly investigated in basic science[16–18]. Having said that there are some treatment regimens based on exposure to extreme temperatures. However, most of these treatments are evidence-based and lack state-of-the-art pre-clinical datasets[19,20]. The understanding of the major impact of the abovementioned stimuli on the translational value of pre-clinical data has only increased in the recent past[21]. As an example, the paper from Orsenigo et al.[22] describes a discovery of a biological effect based on adjusting an independent variable of the microenvironment, using the optical read-out as the dependent variable. By investigating a range of shear stresses, they were able to identify cellular processes otherwise only observable in vivo. Beyond the usage of physical or chemical stimuli as treatment options, they as well must be considered when it comes to data creation in the pre-clinical space. Similarly, any measurement in an assay has an impact on the cell's phenotype. There are several innovative, mainly imaging-based techniques that take this into account and display promising advantages over disruptive or endpoint read-outs[23,24]. This might be one of the reasons for the increase in applied imaging techniques in the field. The deployment of this technology is most advanced in pathohistological applications, clinical as well as pre-clinical[25,26]. The advantage of a truly quantitative analysis of an image enabling the cross-comparison between different experiments has increased the usage of this tool tremendously in the last five to ten years. Other advantages are cloud-based data storage, from which images can be accessed, a fully automated workflow, and the broad applicability of the existing systems across therapeutic areas, basic or translational science. The invention

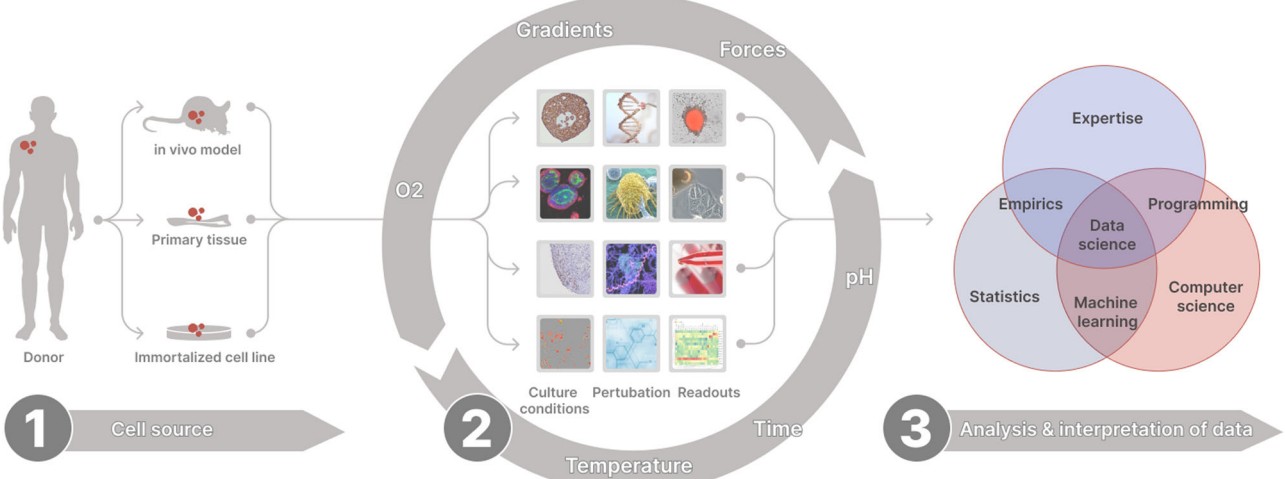

**Fig. 1 Overview of an in vitro workflow.** the design of an in vitro workflow is defined by the choice of cell source (1), the culture conditions, perturbations, and read-outs (2). The analysis and interpretation of the data rely on the seamless integration of statistics, data science, and biological expertise (3). Middle images subtitle from top to bottom: culture conditions: 3D aggregates, 3D organoids, 3D tissue, 2D assays. Perturbation: gene therapy, cell therapy, vaccines, small & large molecules. Readouts: image analysis, mechanics, chemistry, multi-omics.

of high-throughput imaging allowed the application of this technology to living cells. By integrating automation systems that execute image acquisition and analysis, it was possible to improve the throughput without the need to reduce resolution. By these means, imaging became standard in screening assays as well as in the observation of rare events[27]. By the broad implementation of image analysis using deep learning algorithms the value of images as a read-out in cell biology has increased tremendously[28]. One recent example of deep learning in combination with imaging is sensor-extended imaging flow cytometry, which enables high-throughput single-cell analysis and compensates for the technical loss of resolution with a virtual high-resolution image generator[29]. Time is another dimension that was relatively recently added to the cell biology field. Live cell imaging enabled the deconvolution of biological processes over time as well as the increase of accuracy of data by getting independent of pre-defined time points for measurement. The possibility to follow biological processes in real-time opened a new avenue in basic and translational research. However, by applying those new technologies it is important to be aware that also in this case data acquisition is a perturbation per se which has an influence on the dataset[30].

Transitioning from static well-based to flow-based cultures, e.g., microfluidic chips or perfusion models, has further added translational value as demonstrated by multiple groups across many different disciplines[31–33]: the biological advantages of integrating fluidics into an in vitro cell system such as simulation of mechanical forces, circumvention of diffusion and providing controllable culture conditions qualifies this technology for the use in an innovative cell-based system in basic as well as translational research. Beyond that, a fluidics system enables self-referencing sensor readings, realistically displaying the physiological status of the cells, as the flow in the system supports a physiological cell metabolism independent of external stimuli such as passaging or bulk exchange of nutrient-containing media (Fig. 2). Combined with live microscopy imaging, this approach is

the only way to control and monitor the microenvironment of the cultured cells in real-time.

All those technical developments supported by the aim to reduce the use of animals have led to the deployment of highly sophisticated in vitro cell models, so-called microphysiological systems (MPS) that aim to recapitulate human disease. The data generated using MPS helped to understand basic disease mechanisms[34] and in parallel proved to be more predictive than current gold standard animal models of toxicology[35]. MPS enabled the way how we look at cells today. Going forward, we can use the increased knowledge to develop next-generation MPS that will allow us to manipulate and observe the cells simultaneously in a biologically relevant context. Most of the parameters mentioned above are included in the different MPS systems with an emphasis on mimicking organ-specific fluidics such as blood–brain barrier[36], gut[37], or lung[38]. The plethora of different biological components such as cell source or matrices is accompanied by an even greater spectrum of biomechanical features and sensor modules. These different factors that will be combined in many ways to create more sophisticated MPS systems. This process needs input from many different disciplines that must work closely together to lead to a viable, and biologically relevant MPS prototype[2].

## Methods

The challenge of unpredictable project outcomes within a rapidly advancing technology space as described above in biology and biotechnology is similar to the early days of software development. Multiple cell sources, matrices, culture, and measurement devices are available, all with the promise to be most translational-relevant. In analogy to cell biology, within the last 80 years, hundreds of new programming languages appeared, new hardware-enabled novel applications, and displays brought completely new ways of interacting with software and data. Today, software and digital interactions are omnipresent and responsible for some of the biggest recent technological

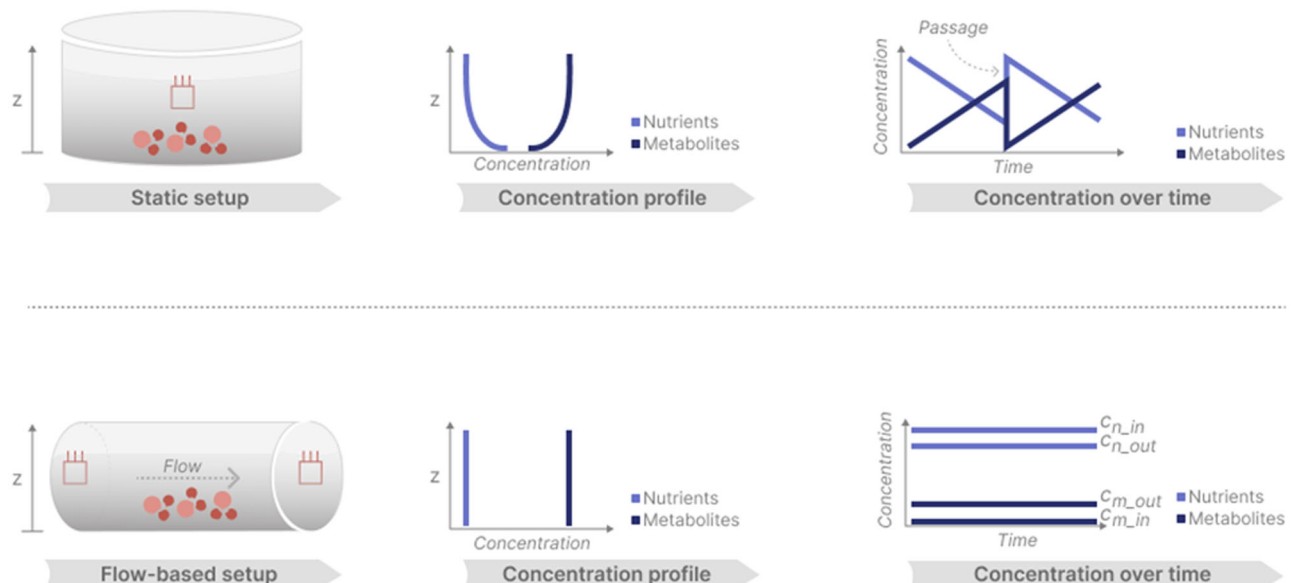

**Fig. 2 Influence of fluidics on cell status.** Influence of fluidics on cell status and its measurability: a static culture within a well is always constrained by diffusion processes that occur due to the cell model consuming nutrients and metabolizing molecules. In a flow-based culture, the concentration gradients due to diffusion are non-existent. An open system technically enables self-referencing sensors to control and monitor different parameters. This difference is qualitatively highlighted in the figure: left panel: cultures are either in a static well or in a flow-based setting; middle panel: cell metabolism leads to concentration gradients along the z-axis in wells, caused by diffusion processes; right panel: passaging abruptly resets the accumulation of metabolites/the concentration of nutrients. In a flow-based culture, the absolute concentrations of nutrients (cn) and metabolites (cm), respectively, can be measured at the inlet (cm/n_in) and outlet (cm/n out) of the cell model with respect to the bulk flow direction.

breakthroughs, including AI-based drug discovery. There is one major difference to cell biology, though: in software development, a monumental effort was taken to deploy development methods that guide programmers as efficiently as possible through the available solution space. Scrum, lean, agile, V-Model, Waterfall, and Wayfaring are such methods, to name a few (Fig. 3). While the technology enables new applications, these new methods enable applying the technology in the most efficient way to create the best possible product for the end user. Similarly, in cell biology, following old pre-software, non-iterative methods of trying to plan everything before executing anything, hinders our ability to develop new biology models most efficiently. In this context, methods describe project management tools that guide the R&D team. For example, Scrum breaks down the complete task into small enough subtasks that a team can implement within a short period of time. This, in return, enables adjusting the path of the development based on new findings, regulations, and especially user insights that come up throughout the development phase[39,40].

Researching the methods in applied projects further brought forward the now widely accepted understanding that iteratively generating and testing prototypes—representations of possible solutions—lead to products that fulfill their purpose more efficiently, are more user-friendly, and overall considered better. The same approach was subsequently adopted in the development of hardware products, where the widespread availability of new tooling, such as 3D printers and laser cutters, reduced the product iteration time in focused R&D teams from months if not years, to days[41].

First reports from MPS developers applying this methodology support our hypothesis that rapid prototyping in an iterative process involving scientists from different disciplines has the potential to increase the translational value of in vitro cell models[42]. Iteration can be applied as a tool at multiple levels: hardware, software, or biology. As an example of a process of integrating new sensors for an organ-on-chip device, Fuchs et al.[2] describe how they need multiple disciplines other than biology before they can iterate the biology itself. This is a step in the right direction. However, the development workflow would be much faster if the iterations could be applied at multiple levels in parallel such as biology, engineering, and data analysis. Bringing in biology as the user of the MPS earlier in the development process of the device has the potential to increase speed and efficiency in

cell biology model development. Sensor-extended imaging aims to enable that process. It allows the application of the above-described product development methods that in turn can make the cell biology model development faster and more efficient, similar to how 3D printing enabled rapid hardware design iterations. The development of an integrated chamber system for live cell microscopy followed that development principle by including multiple prototype-test-iterate rounds, where each iteration focused on a specific problem that gave guidance for the next round. Based on that system it was possible to develop a robust, biocompatible, and easy-to-use device to measure flow-induced shear stress[43].

**Sensor-extended imaging**. The key to advances in science is the ability to observe and record controlled events: one will try in vain to see all the things observed by us in the heavens, as Galileo Galilei stated[44], is as true for astronomy, as it is for particle physics[45]. Yet cell biology has the added complexity of dealing with living organisms that are outside of their physiological context. The need to model a process that for multiple reasons cannot be observed in its natural context is partially adding to the reproducibility crisis in the field[46–48]: Cells are sensitive to much more than just chemical stimuli: any perturbation has an impact on the phenotype of the cell and thereby on the results of the experiment. The advantage of in vitro cell models, to analyze each cell type in isolation, comes with the downside of losing the tissue context without exactly knowing what the influence of this loss might be.

Given that any microenvironmental factors must be considered perturbations and therefore significant factors in an in vitro cell model, there is a clear requirement to control and measure those perturbations. Keeping track of microenvironmental stimuli requires the use of suitable sensors, which are ideally placed outside of the biological specimen to avoid introducing another perturbation.

These measures, in return, enable quality control of the experimental setup. The envisaged accuracy by sensor integration can only be meaningfully achieved by increasing the physical complexity of how cell experiments are conducted. A prerequisite to achieving high versatility and high data accuracy is the change from well- to flow-based assays. In a flow-based assay, the sensor measurements can be implemented without direct perturbation of the cells themselves.

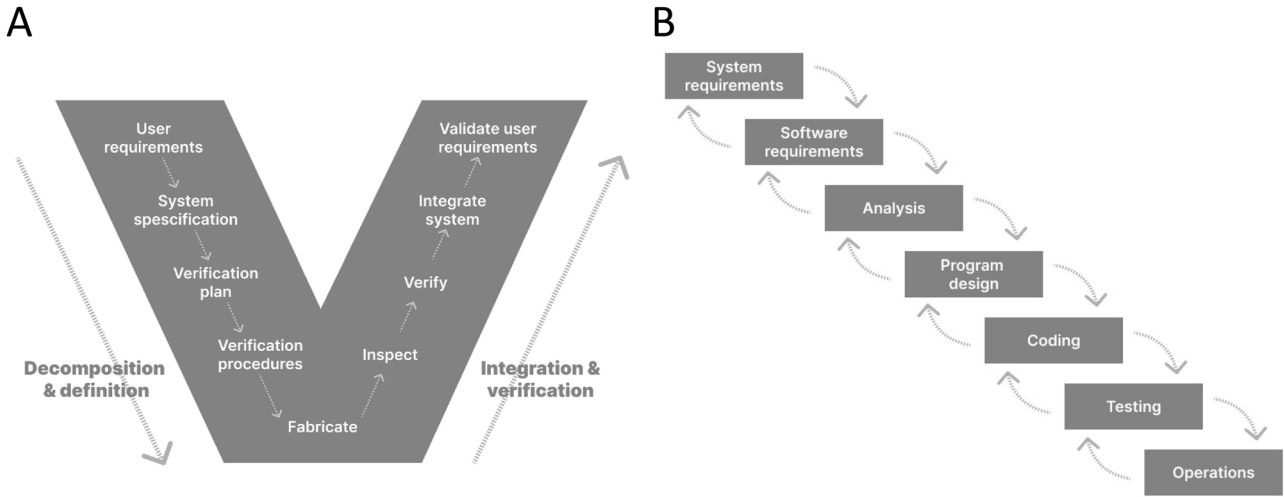

**Fig. 3 Product development models.** Two historically widely applied product development models in software development are the V-model (**A**) and the Waterfall model (**B**). Their benefits and shortcomings triggered the development of more flexible models, such as Agile. Altogether, their applicability for hardware is well established whereas this yet must be proven for assay development models in biotechnology.

The various sensor readings, such as a number of metabolites or oxygen consumption, synchronized with microscopy images bring a level of detail and understanding to cell biology that can significantly increase the validity and translatability of in vitro cell findings to in vivo applications. All these data create a multidimensional image of a time-dependent process that, by necessity, is processed with the support of machine learning and artificial intelligence. Such experimental setups are already used in a wide range of experiments to develop more sophisticated in vitro assays[49], yet only a few of them are so far suitable for a digitally integrated, iterative workflow. Fluidic-based systems like organ-on-a-chip technologies as stand-alone platforms must be combined with novel sensor technologies to speed-up their adoption in drug development, basic science, and precision medicine[35]. We define the above-described sensor-enhanced, interactive imaging workflow *as* sensor-extended imaging, expanding on previous definitions from high-throughput workflows[27] or AI-supported image analytics[14] (Fig. 4).

One-way to drive adoption is the design of an experimental setup with integrated sensors, a fluidics compartment that allows application as well as retrieval of samples, microscopic imaging, and a cloud-based software system to process the data in real-time. The implementation of such a workflow, based on the learnings from software and hardware product development processes, is a dynamic, multidisciplinary, and iterative approach. It enables the user to dynamically induce controlled perturbations to a cell model and in parallel record the results in real-time. There are three main advantages to this approach: (a) recording complete events creates a histogram of what a cell has experienced, provides valuable information for backtracking error sources, (b) the dynamic access to the cell model enables rapid testing and iterating hypothesis-generating experiments and (c) training of ML-models combining imaging with sensor data in real-time increases the translational value of the platform. The sensor-supported real-time control enables precise, programmable perturbations to establish a physiological microenvironment in the cell compartment. Similarly, the recording of sensor readings, as well as microscopy images gives real-time feedback of how the perturbations affect the cell models. This enables the processing of fundamentally new research questions, not only when it comes to perturbations as a function of time (instead of using time indirectly through pre-defined time points for measurement), but it also allows using new recognizable events by the sensors or fluorescent images as a trigger for applying perturbations (i.e., give a drug when certain cell confluence is reached or a biomarker is recognized). Taken together, the transition from imaging to sensor-extended imaging can be characterized by the addition of different measures with programmable actuation plus the application of machine learning to set the data into context.

**Perspective**. The current goal of biology is to observe and understand as you cannot dissect and manipulate a patient but must copy one either in an animal or in a petri dish. Due to the complexity of nature, none of the current systems capture all aspects of the biology it aims to mimic. The science in the field has advanced in a way that it is no longer sufficient to manipulate and measure the effect of this manipulation but to decipher the environmental changes a cell is exposed to by those measurements. This tracking can only be feasible with the help of digital sensors and tools. This fundamental difference of recording and manipulating the cell perturbations in a digital way changes the way we can start looking at biology experiments—not anymore as a one-way street of observing the cell, but as a real-time multivariable interaction between the living and the programmable perturbations. This is described as a sensor-extended imaging workflow that enables faster iterations of the biology instead of the engineering part of the MPS. Similar to the invention of 3D printers enabling innovation in hardware, new tools, in this case, the sensor-extended imaging, will enable explorative, iterative, and controlled workflows in cell biology. As in the advent of iPSC, which revolutionized the use of human cells in vitro, it will be possible to design very rapidly MPS that are fit for purpose and replicate a specific aspect of (patho)physiological status of an organ or organ system[50]. For MPS to transform the applied sciences, there are practical challenges to be solved, including usability, availability,

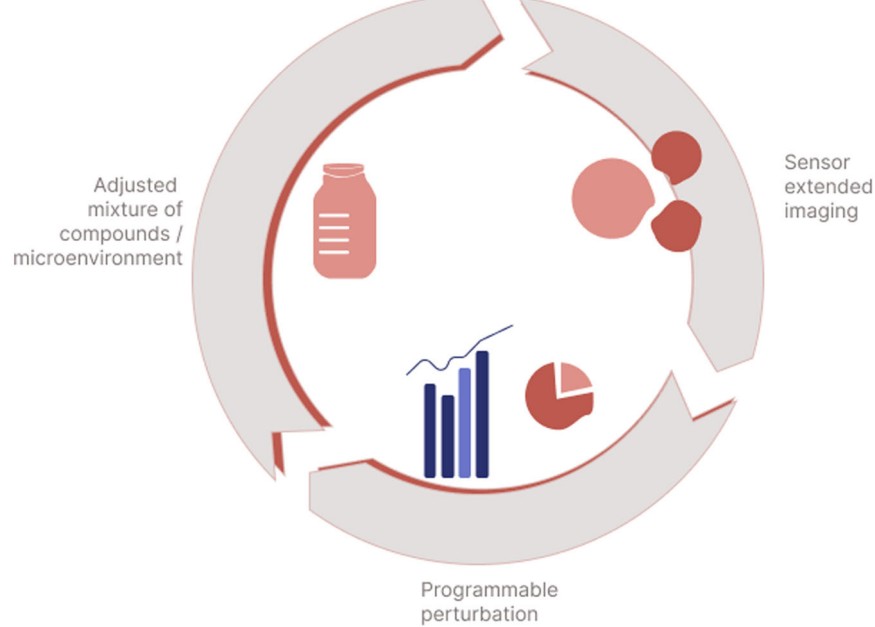

**Fig. 4 Sensor-extended imaging workflow for in vitro cell models.** A sensor-extended imaging workflow allows for generating data from a cell experiment and using these data in parallel as control parameters for future perturbations. This is enabled by an iterative process adjusting the supply of nutrients and compounds, programmable perturbations, and sensor-extended imaging measurements to each other.

standardization, sterilization, to mention a few, also discussed by Fuchs et al.[2]. We believe that those challenges can be solved by developing chips, sensors, imaging solutions, and fluidics interfaces separately in a modular fashion fit for purpose. Applying the sensor-extended imaging as the underlying workflow for the biology cell model development, enables the iterative development urgently needed while controlling all perturbations. Thereby, truly fit-for-purpose cell biology models will be made available to the scientific community.

**Reporting summary**. Further information on research design is available in the Nature Portfolio Reporting Summary linked to this article.

# Data availability

We did not analyze or generate any datasets, because our work proceeds within a theoretical approach.

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

## Author contributions

J.S. provided expertise in cell biology, wrote, and edited the manuscript, H.S. provided expertise in IoT, wrote, and edited the manuscript, C.K. provided expertise in engineering, wrote, and reviewed the manuscript.

## Competing interests

The authors declare no competing interests.

## Additional information

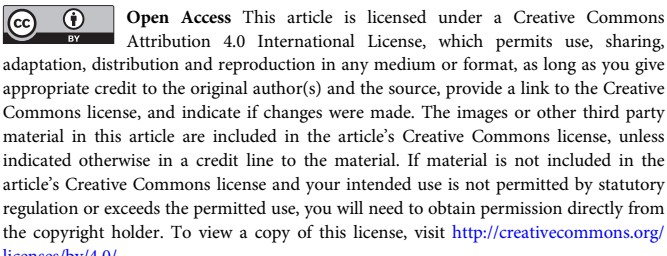

