## [Peer Review File · Communications Biology]

Reviewers' comments:

Reviewer #1 (Remarks to the Author):

In this work, Schueler et al., by referring to the recent FDA modernization act, allude to the growing need for design and development of bespoke in vitro cell assays. Then, by drawing an analogy between approaches adopted in product development, software/hardware engineering and biotechnology they conclude that employing artificial intelligence, machine learning, internet of things, product development processes, prototyping and including multidisciplinary expertise are of crucial importance for the design and development of the in vitro assays. Eventually, the authors propose deep imaging, i.e., incorporation of sensors and high-throughput imaging in an interactive manner, as a base for the next generation of cell based in-vitro assays. The work covers a wide range of topics, as it is a necessity in the multidisciplinary approach, however there are several concerns about this review.

- The title suggests that the main message of the manuscript is related to "deep imaging" which is defined by the authors as "the combination of image acquisition and sensor measurement". However, the manuscript lacks any information about microelectromechanical systems (MEMS) and microoptoelectromechanical systems (MOEMS) which have been used as successful examples of integration of sensors and microscopy techniques. Additionally, different types of sensors, their applications and integrations with the in vitro systems have not been discussed in the manuscript. Although the authors have recognized the importance of fluidic based systems, they have not provided any information about successful examples of microfluidic-based cell assays.
- The quality of the figures is very low, and they are not discussed properly in the text.
- Application of machine learning, artificial intelligence, and internet of things has not been discussed deeply in the context of cell-based assays.
- The manuscript has several sentences whose relevance is not clear, for instance: "To drive this implementation, standardization will be key, and best be achieved in the framework of public private Partnerships".
- There are some typos and punctuation mistakes in the manuscript.

Reviewer #2 (Remarks to the Author):

In "Deep imaging for accelerating decision making in basic and applied cell biology" the authors discuss the need to accelerate the iteration of microphysiological systems, using the software development cycle as an analogy. They argue that integrating sensor data with images, so called "Deep Imaging" is a means to acquiring more data without perturbing cells.

While the manuscript raises some points on the importance of iteration, I found that the main thesis of the manuscript is too vague and lacks insight.

- The term "Deep Imaging" is used already to represent imaging deep into tissue, and seems to be a very confusing term.
- Integrating sensors with images is already almost universally employed in organ on chip systems, and it is not a particularly new insight that multi-modal measurements have important applications. In fact, this topic is already well covered in reviews such as: [10.1021/acsbiomaterials.0c01110](https://doi.org/10.1021/acsbiomaterials.0c01110)
- There are many vague sentences whose premise I do not consider correct.
- "To achieve [modulation and measurement in real-time], a fluidic based system is mandatory". Such measurements could be made in a well-plate.
- "there is a lack of interdisciplinary work between biology and engineering on many levels". The whole field of bioengineering very actively works in this area. Biologists have adopted incredible new

engineering and computational methods in the past decade.

- "The attempt to add a digital component to a scientific process has created inflexible, expensive, and complicated systems". This is a confusing statement.

- "Software is unique in the way that text and symbols compile into executable applications within seconds at virtually no cost..." I don't think this discussion is particularly relevant to integrating sensors and images.

- The figures are not particularly insightful. Figure 2, in particular, is very vague and does not teach me how integrating sensors and image data is helpful.

If submitting elsewhere, I would suggest altering this discussion to provide a more wholistic and in-depth study on integrated sensing and imaging.

In summary, in light of the vague, imprecise, and occasionally misleading statements throughout this manuscript, I do not recommend its publication.

Reviewer #3 (Remarks to the Author):

Comments on Schueler et al., "Deep imaging for accelerated decision making in basic and applied cell biology"

Major comments:

1) The article is stated to be a review ("This review aims...), but it is in fact a perspective piece that is divided between commentary on how to improve/speed product development in biotechnology (Sections 1-3 & 6) and how/why recording environmental variables is important for the interpretability/reproducibility of cell culture experiments (Sections 4,5). The sections on the importance of environmental variables are interesting and relevant to the typical academic researcher. The product development sections feel better suited for a biotechnology- or pharma-oriented business journal. The article would benefit from being split in two, one article focusing on each of these two major topics.

2) The sentence structure is often difficult to follow and should be re-written more concisely, avoiding asides and unnecessarily flowery language. A few examples:

Authors: "Since the first experiments involving animal cells in isolation approximately 100 years ago, the medical need especially in virology and oncology has initiated the development of a plethora of different in vitro technologies that aim to mimic the human diseased or healthy archetype." (43 words)
Why not: "Since the dawn of cell culture, many in vitro technologies have been developed to mimic human health or disease states." (20 words)

Authors: "There are several research fields cell culture systems are important contributors, such as a) model specific diseases to expand our knowledge of the disease and ultimately develop a cure, b) to understand the physiological requirements of bespoke cell types and along those lines be able to determine negative and positive effects of perturbations on those cells, c) as basis of manufacturing systems for therapeutic proteins, cell and gene therapies." (69 words)

Why not: "Cell cultures systems are important to many research fields, being commonly used for pre-clinical disease modeling, basic cell physiology research, and for manufacturing biologics." (24 words)

Minor Comments:

The authors coin the phrase "deep imaging" to mean a collection of image data as well as sensor data

defining the cell culture environment. Unfortunately, the phrase "deep imaging" is already in common use where it typically means imaging far into a tissue or imaging far from a coverslip. To avoid confusion, when this phrase is used in the article title (where it is undefined), it should be enclosed in quotations to indicate a specialized meaning. Can a more descriptive phrase be found (sensor fusion imaging?)?

Given the author's conclusions imply a need for products that the author's sell, should this article be labeled as an advertising feature?

CRL GmbH • Am Flughafen 12-14 • D-79108 Freiburg • Germany

Editorial Office
Nature Communications Biology

Julia Schüler, DVM PhD
therapeutic area lead oncology

phone: +49(0)761 51559-40
fax: +49(0)761 51559-55
email: Julia.schueler@crl.com

date: 04.10.2023

Dear Dr Chong and Dr Fritzsche,

thank you very much for the editorial and reviewer comments to our manuscript “Deep imaging for accelerating decision making in basic and applied cell biology”. Please find below the point-by-point answer’s to the reviewer comments.

Sincerely in the name of all co-authors,

Julia Schüler, DVM PHD
research director, therapeutic area lead oncology

Reviewer #1 (Remarks to the Author):

In this work, Schueler et al., by referring to the recent FDA modernization act, allude to the growing need for design and development of bespoke in vitro cell assays. Then, by drawing an analogy between approaches adopted in product development, software/hardware engineering and biotechnology they conclude that employing artificial intelligence, machine learning, internet of things, product development processes, prototyping and including multidisciplinary expertise are of crucial importance for the design and development of the in vitro assays. Eventually, the authors propose deep imaging, i.e., incorporation of sensors and high-throughput imaging in an interactive manner, as a base for the next generation of cell based in-vitro assays. The work covers a wide range of topics, as it is a necessity in the multidisciplinary approach, however there are several concerns about this review.

- The title suggests that the main message of the manuscript is related to “deep imaging” which is defined by the authors as “the combination of image acquisition and sensor measurement”. However, the manuscript lacks any information about microelectromechanical systems (MEMS) and micro-opto-electromechanical systems (MOEMS) which have been used as successful examples of integration of sensors and microscopy techniques.

The aim of the manuscript (perspective) is to propose a workflow/method applying to the development of different platforms rather than landscaping the different existing technologies. We rephrased large parts of the manuscript to make this point unambiguous.

Additionally, different types of sensors, their applications and integrations with the in vitro systems have not been discussed in the manuscript. Although the authors have recognized the importance of fluidic based systems, they have not provided any information about successful examples of microfluidic-based cell assays.

As we changed the format of the manuscript towards a Perspective, we do not aim to cover the full range of all available systems currently in the field. However, we elaborated more in detail, on which aspects in biology those microphysiological systems had a major impact: “The trend towards more human models supported by the tendency to reduce the use of animals has led to the deployment of highly sophisticated in vitro cell models, so called microphysiological systems (MPS) that aim to recapitulate the human disease. The data generated using MPS helped to understand basic disease mechanisms and in parallel proved to be more predictive than current gold standard animal models of toxicology. MPS enabled the way how we look at cells today. Going forward, we can use the increased knowledge to develop next generation MPS that will allow to manipulate and observe the cells simultaneously in a biological relevant context.”

- The quality of the figures is very low, and they are not discussed properly in the text.

We have redesigned the figures and hope that the increased quality fulfills the requirement of the journal.

- Application of machine learning, artificial intelligence, and internet of things has not been discussed deeply in the context of cell-based assays.

As we have rephrased large parts of the manuscript and focus the revision more specifically on the development process of MPS, we excluded that aspect. To make the perspective more concise we decided not to focus on the digital component but on the interaction between sensors and cells.

- The manuscript has several sentences whose relevance is not clear, for instance: “To drive this implementation, standardization will be key, and best be achieved in the framework of public private Partnerships”.

We hope the revised more concise version of the manuscript delivers a more precise description of the topic and possible implications of our suggested workflow on the field of MPS.

page 2 of 5

-There are some typos and punctuation mistakes in the manuscript.

We thoroughly reviewed and edited the text and hope that we eliminated all typos.

Reviewer #2 (Remarks to the Author):

In "Deep imaging for accelerating decision making in basic and applied cell biology" the authors discuss the need to accelerate the iteration of microphysiological systems, using the software development cycle as an analogy. They argue that integrating sensor data with images, so called "Deep Imaging" is a means to acquiring more data without perturbing cells.

While the manuscript raises some points on the importance of iteration, I found that the main thesis of the manuscript is too vague and lacks insight.

We hope the revised more concise version of the manuscript delivers a more precise description of the topic and possible implications of our suggested workflow on the field of MPS.

- The term "Deep Imaging" is used already to represent imaging deep into tissue, and seems to be a very confusing term.

To make our statements unambiguous, we changed the term to "sensor extended imaging"

- Integrating sensors with images is already almost universally employed in organ on chip systems, and it is not a particularly new insight that multi-modal measurements have important applications. In fact, this topic is already well covered in reviews such as: 10.1021/acsbiomaterials.0c01110

We value the comment of the reviewer. However, the aim of the manuscript, now in the format of a perspective, is to define a possible solution to some of the challenges described by Fuchs et al in the above-mentioned manuscript. In the revised version we intend to describe this point unmistakably. Another crucial difference is that in their definition of integration with images is done after-the-fact, with fixed samples, not live as we define in the sensor extended imaging. They also mention this in their manuscript by stating: "Label-free and continuous real-time analysis of cell viability parameters remains one of the most important unresolved technical challenges in advancing OoC models." Post experiment, stained imaging, does not offer the opportunity to iterate with similar rate as with live imaging.

- There are many vague sentences whose premise I do not consider correct.

- "To achieve [modulation and measurement in real-time], a fluidic based system is mandatory". Such measurements could be made in a well-plate.

We rephased the sentence to make it more specific: To achieve this goal in advanced in vitro cell model, a fluidic based system is mandatory. The fluidic system gives a possibility to measure without introducing a perturbation to the cell model and enables a data acquisition completely uncoupled from the perturbation to be evaluated in the respective experiment.

- "there is a lack of interdisciplinary work between biology and engineering on many levels". The whole field of bioengineering very actively works in this area. Biologists have adopted incredible new engineering and computational methods in the past decade.

We rephased the complete section. The most relevant statement can be found in the chapter "methods"

- "The attempt to add a digital component to a scientific process has created inflexible, expensive, and complicated systems". This is a confusing statement.

To make the perspective more concise we decided not to focus on the digital component but on the interaction between sensors and cells.

- "Software is unique in the way that text and symbols compile into executable applications within seconds at virtually no cost..." I don't think this discussion is particularly relevant to integrating sensors and images.

We omitted the sentence from the manuscript and the discussion was rephrased to better fit the context: how to translate already developed software and hardware product development practices to cell biology through integrating sensor extended image workflow.

- The figures are not particularly insightful. Figure 2, in particular, is very vague and does not teach me how integrating sensors and image data is helpful.

We have rephrased the section and re-ordered the manuscript to better explain the link between process development models and MPS systems.

If submitting elsewhere, I would suggest altering this discussion to provide a more holistic and in-depth study on integrated sensing and imaging.

In summary, in light of the vague, imprecise, and occasionally misleading statements throughout this manuscript, I do not recommend its publication.

We hope that the revised version of the manuscript delivers a more distinct message underpinned by a clear argumentation.

Reviewer #3 (Remarks to the Author):

Comments on Schueler et al., "Deep imaging for accelerated decision making in basic and applied cell biology"

Major comments:

1) The article is stated to be a review ("This review aims...), but it is in fact a perspective piece that is divided between commentary on how to improve/speed product development in biotechnology (Sections 1-3 & 6) and how/why recording environmental variables is important for the interpretability/reproducibility of cell culture experiments (Sections 4,5). The sections on the importance of environmental variables are interesting and relevant to the typical academic researcher. The product development sections feel better suited for a biotechnology- or pharma-oriented business journal. The article would benefit from being split in two, one article focusing on each of these two major topics.

Following the reviewer's recommendation, we rephrased the manuscript as a Perspective and focused on the aspect relevant for the research community.

2) The sentence structure is often difficult to follow and should be re-written more concisely, avoiding asides and unnecessarily flowery language. A few examples:

Authors: "Since the first experiments involving animal cells in isolation approximately 100 years ago, the medical need especially in virology and oncology has initiated the development of a plethora of different in vitro technologies that aim to mimic the human diseased or healthy archetype." (43 words)

Why not: "Since the dawn of cell culture, many in vitro technologies have been developed to mimic human health or disease states." (20 words)

This part of the manuscript was re-written to adapt to the comment above.

Authors: "There are several research fields cell culture systems are important contributors, such as a) model specific diseases to expand our knowledge of the disease and ultimately develop a cure, b) to understand the physiological requirements of bespoke cell types and along those lines be able to

page 4 of 5

determine negative and positive effects of perturbations on those cells, c) as basis of manufacturing systems for therapeutic proteins, cell and gene therapies.” (69 words)

Why not: “Cell cultures systems are important to many research fields, being commonly used for pre-clinical disease modeling, basic cell physiology research, and for manufacturing biologics.” (24 words)

This part of the manuscript was re-written to adapt to the comment above.

Minor Comments:

The authors coin the phrase “deep imaging” to mean a collection of image data as well as sensor data defining the cell culture environment. Unfortunately, the phrase “deep imaging” is already in common use where it typically means imaging far into a tissue or imaging far from a coverslip. To avoid confusion, when this phrase is used in the article title (where it is undefined), it should be enclosed in quotations to indicate a specialized meaning. Can a more descriptive phrase be found (sensor fusion imaging?)?

To make our statements unambiguous, we changed the term to “sensor extended imaging”

Given the author’s conclusions imply a need for products that the author’s sell, should this article be labeled as an advertising feature?

The aim of the manuscript is not to imply a need for a specific product but suggests a general workflow when developing a product in the biotechnology realm. We hope with the revised manuscript as a perspective we made this message unambiguous.

Reviewers' comments:

Reviewer #1 (Remarks to the Author):

The authors have adequately addressed my concerns. I can now endorse the manuscript for publication as a Perspective.

Reviewer #2 (Remarks to the Author):

The revised manuscript is now written as a perspective and the language, clarity, and conciseness has been improved. That said, I believe that my main critique that the Perspective is overall too vague, and lacks an interesting or novel insight. The figures also still do not provide any clarity.

As noted by the authors, Fuchs et al. is one example of a review that offers a more complete and in-depth review of real-time sensors in MPS and even offers a perspective on multimodal sensing and its value. The stated intent of the authors is "to define a possible solution to some of the challenges" in integrating live imaging into an MPS workflow. I truthfully don't consider their insights here to be particularly novel or interesting. One specific example:

"One way to drive adoption is the design of an experimental setup with integrated sensors, a fluidics compartment that allows application as well as retrieval of samples, microscopic imaging, and a cloud based software system to process the data in real time."

Raspberry Pis and other low cost systems have been used as cloud servers for closed-loop remote imaging and remote control for over a decade, e.g. doi:10.1039/c2lc41000a, doi:10.1109/SIITME.2017.8259880 among many others.

I unfortunately do not consider this Perspective to be particularly valuable for the MPS or broader biology fields.

Reviewer #3 (Remarks to the Author):

The authors have extensively re-written the manuscript as a Perspective. They outline how they believe the integration of fluidics (to precisely control media composition), sensors (to measure environmental variables), and other metadata (cell sourcing etc.) can speed discovery and aid reproducibility. Indeed, all these considerations are very important but yet typically missing from current in vitro experiments.

I suspect these considerations are usually ignored due to the complexity of their implementation. It would be interesting for biologists if the authors could present some concrete, real-world examples where sensor extended imaging has been fruitfully applied, along with a description of major hurdles that were encountered. (While the authors allude to 'sensor extending imaging flow cytometry' on lines 101-102, in fact this reference only describes application of deep learning for image restoration. There is no integration of environmental sensor data or other metadata).

Similarly, in the section on software development methods (lines 135-152), it may be true that biologists can learn to be more efficient problem solvers, but without a concrete example of success it is unclear to me how/why methods that speed coding should necessarily also speed biological discovery.

The ideas presented are good ones but without more concrete examples it will be difficult to convince biologists that these ideas can actually be usefully applied in practice.

CRL GmbH • Am Flughafen 12-14 • D-79108 Freiburg • Germany

Editorial Office
Nature Communications Biology

Julia Schüler, DVM PhD
therapeutic area lead oncology

phone: +49(0)761 51559-40
fax: +49(0)761 51559-55
email: Julia.schueler@crl.com

date: 30.12.2023

Dear Dr Chong, Dr Fritzsche and Dr Breuer

thank you very much for the reviewer comments to our manuscript "Sensor extended imaging workflow for creating fit for purpose models in basic and applied cell biology". Please find below the point-by-point answer's to the reviewer comments.

Sincerely in the name of all co-authors,

Julia Schüler, DVM PHD
research director, therapeutic area lead oncology

Reviewers' comments:

Reviewer #1 (Remarks to the Author):

The authors have adequately addressed my concerns. I can now endorse the manuscript for publication as a Perspective.

Reviewer #2 (Remarks to the Author):

The revised manuscript is now written as a perspective and the language, clarity, and conciseness has been improved. That said, I believe that my main critique that the Perspective is overall too vague, and lacks an interesting or novel insight. The figures also still do not provide any clarity.

As noted by the authors, Fuchs et al. is one example of a review that offers a more complete and in-depth review of real-time sensors in MPS and even offers a perspective on multimodal sensing and its value. The stated intent of the authors is "to define a possible solution to some of the challenges" in integrating live imaging into an MPS workflow. I truthfully don't consider their insights here to be particularly novel or interesting. One specific example:

"One way to drive adoption is the design of an experimental setup with integrated sensors, a fluidics compartment that allows application as well as retrieval of samples, microscopic imaging, and a cloud based software system to process the data in real time."

Raspberry Pis and other low cost systems have been used as cloud servers for closed-loop remote imaging and remote control for over a decade, e.g. doi:10.1039/c2lc41000a, doi:10.1109/SIITME.2017.8259880 among many others.

I unfortunately do not consider this Perspective to be particularly valuable for the MPS or broader biology fields.

Our perspective approaches the challenge of developing and deploying MPSs from a product development mindset as pioneered by software developers and their methods – we agree with Fuchs et al. that real-time sensing can define a solution to some of the challenges, yet we aim to also answer *how* the solution can be used.

We agree that the fact that RPIs and cloud infrastructure is used in biotech is not novel yet at the same time this is *not* about passive imaging, but about active control and monitoring. The fact that one can monitor *and* control multiple variables in real time (as in: not just passive optical readouts but also actively controlled microenvironmental variables) to map out a multidimensional solution space to a problem is, to our knowledge across research and industrial applications, not a previously described and not commonly used approach in live cell biology.

The design and implementation of in vitro experiments today does not take into account environmental influences given by the geographic or infrastructural conditions of the executing laboratory. Achieving truly precise, predictive, and repeatable MPS for translating laboratory efforts to patient outcomes requires many more dimensions of control than just closed-loop remote imaging – our aim with this perspective is to highlight why and how this is the case from an R&D perspective.

Reviewer #3 (Remarks to the Author):

page 2 of 3

The authors have extensively re-written the manuscript as a Perspective. They outline how they believe the integration of fluidics (to precisely control media composition), sensors (to measure environmental variables), and other metadata (cell sourcing etc.) can speed discovery and aid reproducibility. Indeed, all these considerations are very important but yet typically missing from current in vitro experiments.

I suspect these considerations are usually ignored due to the complexity of their implementation. It would be interesting for biologists if the authors could present some concrete, real-world examples where sensor extended imaging has been fruitfully applied, along with a description of major hurdles that were encountered. (While the authors allude to 'sensor extending imaging flow cytometry' on lines 101-102, in fact this reference only describes application of deep learning for image restoration. There is no integration of environmental sensor data or other metadata).

Similarly, in the section on software development methods (lines 135-152), it may be true that biologists can learn to be more efficient problem solvers, but without a concrete example of success it is unclear to me how/why methods that speed coding should necessarily also speed biological discovery.

The ideas presented are good ones but without more concrete examples it will be difficult to convince biologists that these ideas can actually be usefully applied in practice.

We are fully in line with the reviewer's comment that the key is, indeed, the complexity of the implementation. We added an example of purely sensor extended imaging by Orsenigo, F., et al, <https://pubmed.ncbi.nlm.nih.gov/23169049/> where they described a discovery of a biological effect based on adjusting an independent variable of the microenvironment, using the optical readout as the dependent variable. By investigating a range of shear stresses, they were able to identify cellular processes otherwise only observable in vivo. Their complex setup would be extremely cumbersome to reproduce (A custom-built flow chamber consisting of two parallel plates made of polymethyl-methacrylate). Adding controls always means that meaningful measurements must be taken, and that the according independent variable can be actively adjusted. In <https://www.frontiersin.org/articles/10.3389/fbioe.2019.00091/full> we described the steps to get temperature control of flowing medium implemented within a meaningful envelope that fits on a standard microscope stage significantly lessening the hurdles of setting up a sensor extended imaging experiment in cell biology. Another example applying this concept to an in vivo setting was described by J. Blindheim (<https://ntnuopen.ntnu.no/ntnu-xmlui/handle/11250/2615420>) who developed a completely novel experiment setup to investigate two photon microscopy in mice. The definite proof that this approach is faster and better can only be achieved in a prospective head-to-head comparison. However, we hope that providing some real-world examples of the advantages of this concept will convince the audience of usefulness of this perspective.

REVIEWERS' COMMENTS:

Reviewer #3 (Remarks to the Author):

My main concerns have been addressed. I leave it to the editors to decide if the manuscript is of sufficient interest and impact for the journal.

CRL GmbH • Am Flughafen 12-14 • D-79108 Freiburg • Germany

Editorial Office
Nature Communications Biology

Julia Schüler, DVM PhD
therapeutic area lead oncology

phone: +49(0)761 51559-40
fax: +49(0)761 51559-55
email: Julia.schueler@crl.com

date: 17.01.2024

Dear Dr Fritzsche and Dr Breuer

thank you very much for the reviewer comments to our manuscript "Sensor extended imaging workflow for creating fit for purpose models in basic and applied cell biology". Please find below the point-by-point answer's to the reviewer comments.

Sincerely in the name of all co-authors,

Julia Schüler, DVM PHD
research director, therapeutic area lead oncology

Reviewers' comments:

Reviewer #3 (Remarks to the Author):

My main concerns have been addressed. I leave it to the editors to decide if the manuscript is of sufficient interest and impact for the journal.

We thank the reviewer for the constructive reviewing process.